



# A two-stage Bayesian multi-model framework for improving multidimensional drought risk projections over China

Boen Zhang[1], Shuo Wang[1,2], and Jinxin Zhu[1]

[1]Department of Land Surveying and Geo-Informatics, The Hong Kong Polytechnic University, Hong Kong, China

[2]The Hong Kong Polytechnic University Shenzhen Research Institute, Shenzhen, China

*Correspondence to*: Shuo Wang (shuo.s.wang@polyu.edu.hk)

**Abstract.** Understanding future drought risk is a prerequisite for developing climate change adaptation strategies and for enhancing disaster resilience. In this study, we develop multi-model probabilistic projections of multidimensional drought risks under two representative emission scenarios (RCP4.5 and RCP8.5) through a copula-based Bayesian framework. An ensemble of five regional climate simulations, including four from the CORDEX East Asia experiment and one from the Providing REgional Climate Impacts for Studies (PRECIS) simulation, is used to project future changes in hydroclimatic regimes over China. A new Bayesian copula approach is introduced to uncover underlying interactions of drought characteristics and associated uncertainties over 10 climate divisions of China. The proposed Bayesian framework explicitly addresses the cascade of uncertainty in high-resolution projections of multidimensional drought risks. Our findings reveal that precipitation and potential evapotranspiration (PET) are projected to increase for most areas of China, while increasing radiative forcing is expected to amplify the increase in PET but does not cause significant changes in the precipitation projection. In addition, the drought duration and severity are projected to substantially increase for most areas of China. The estimated drought risks in China are expected to become more than double under both emission scenarios. The extreme droughts are projected to intensify in terms of frequency and associated risks as the radiative forcing increases.

## 1 Introduction

Droughts, one of the costliest and most widespread natural hazards, have caused massive economic losses, environmental degradation and even loss of human life around the world (Dai, 2013; Samaniego et al., 2018; Su et al., 2018). For example, a severe and prolonged drought episode during 2009 and 2010 affected millions of people and livestock in northern and southwestern China with billions of dollars in economic losses (Barriopedro et al., 2012). Considering the substantial impacts of droughts and the indisputable fact of global warming, assessing the evolution of drought risks in a changing climate has received considerable attention in recent decades (Chen et al., 2020; Cook et al., 2016; Prudhomme et al., 2014).



Since the coarse-resolution global climate models (GCMs) fail to obtain regional or local-scale hydroclimatic characteristics, regional climate models (RCMs) driven by GCM outputs or reanalysis data have been extensively used for regional climate impact assessments (Asadi Zarch et al., 2015; F Giorgi et al., 2009; Filippo Giorgi, 2019; Van Huijgevoort et

al., 2014; Russo et al., 2013; Sheffield and Wood, 2008; Zhu et al., 2019). A reliable projection of drought hazard changes requires multi-model simulations of hydroclimatic regimes. The common practice is to calculate the arithmetic ensemble mean (AEM) of drought variables (e.g., precipitation) derived from multiple RCMs for drought characterization and projection, which has been proven to enable more accurate simulation than a single climate model (Rajsekhar and Gorelick, 2017). For example, Lee et al. (2019) found that the severity of future droughts in Korea was expected to deteriorate with enhanced

confidence by using multi-model ensemble projections. Senatore et al. (2019) used multiple RCM datasets from the Coordinated Regional Downscaling Experiment (CORDEX) program to project drought occurrence in Iran. Since the AEM method cannot completely avoid the errors of individual models, Bayesian model averaging (BMA) as a promising method has been widely used for multi-model hydroclimate simulations in recent years (Ahmadalipour et al., 2018; Duan et al., 2007; Madadgar and Moradkhani, 2014; Raftery et al., 1997, 2005; Zhang et al., 2016; Olson et al., 2016, 2018; Shin et al., 2019).

For example, Min and Hense (2006) found that BMA outperformed the AEM method in simulating global mean surface temperature. Duan and Phillips (2010) combined multiple GCM outputs using BMA to project future changes in continental temperature and precipitation. Yang et al. (2011) analyzed the spatiotemporal evolution in climate extremes in the Tarim River Basin located over Central Asia based on BMA and five GCMs. Although BMA has been used to improve climate simulations in previous studies, little effort has been directed towards applying BMA to projecting future drought characteristics.

BMA relies on accurate parameter estimates of the individual climate models in the ensemble, which is typically realized using the Expectation-Maximization (EM) algorithm (Ahmadalipour et al., 2018; Raftery et al., 1997; Yang et al., 2011; Zhang et al., 2016). Such a practice, however, fails to guarantee the global convergence of BMA weights and variances as well as cannot provide the corresponding uncertainty information (Vrugt, Diks, et al., 2008; Vrugt, 2016). Although the EM algorithm provides plausible climate simulations, ignoring the underlying uncertainty diminishes the reliability of the assessment of

hydrological extremes. There has also been a general agreement in the hydrologic community that probabilistic hydrological predictions outperform deterministic predictions (Ramos et al., 2013; Wang et al., 2018). Therefore, it is desired to improve climate-induced drought risk assessments through probabilistic BMA-based climate projections (Wang et al., 2019; Zhang et al., 2019).

In addition to projections of drought hazard changes, drought frequency analysis is widely used in drought resistance

planning and drought vulnerability assessment (Borgomeo et al., 2015; Hao et al., 2017; Hao and AghaKouchak, 2013; Liu et al., 2016; Seager et al., 2015; Williams et al., 2015). The conventional univariate frequency analysis is considered unreliable since a single variable is insufficient to characterize drought risks (Kam et al., 2014). The multivariate frequency analysis has thus attracted increasing attention, which takes into account the interrelationships of drought characteristics (i.e., drought severity, spatial extent, and duration, etc.) (Ayantobo et al., 2018; Carvalho and Wang, 2019; Maity et al., 2013). Copula has

gained remarkable success in multivariate drought analysis owing to its flexibility in capturing the complicated dependencies





between drought characteristics regardless of their marginal distributions (AghaKouchak et al., 2014; Ganguli and Reddy, 2014; Masud et al., 2017; Sadegh et al., 2018; Salvadori and De Michele, 2004; Salvadori et al., 2016). For example, Xu et al. (2015) considered the spatial extent of droughts in the copula-based multivariate drought frequency analysis in Southwest China. Zhang et al. (2019) used copula and the convection-permitting climate simulations to assess climate change impacts on

the multivariate drought evolution over South Central Texas. One of the most important variables derived by multivariate drought frequency analysis is the drought return period, which represents the average time between drought episodes and thus quantifies drought risks (Kwon and Lall, 2016; Masud et al., 2015). For example, AghaKouchak et al. (2014) indicated that the 2014 California drought can be a 200-year extreme event when considering the combined effects of low-precipitation and high-temperature conditions. Liu et al. (2016) concluded that the average multivariate return period of extreme droughts in

China during 1961−2013 was 42.1 years. Previous studies, however, fail to guarantee the global optimization of copula parameters by using the frequentist approach, leading to a potential bias in drought return periods. Another limitation of previous studies is that they fail to quantify the underlying uncertainties of copula parameters. Such uncertainty is considerably large since the samples of drought episodes are typically limited, and ignoring the uncertainty diminishes the scientific credibility in drought assessments (De Michele et al., 2013; Sadegh et al., 2017). Therefore, it is necessary to explicitly address

the uncertainty in copula-based drought risk assessments for advancing our understanding of complex mechanisms and potential impacts of droughts.

The objective of this study is to develop a two-stage Bayesian multi-model framework to improve the multidimensional drought risk assessments in a changing climate. Specifically, an ensemble of five regional climate simulations, including four from the CORDEX East Asia experiment and one from the Providing REgional Climate Impacts for Studies (PRECIS)

simulation will be used to improve the performance of climate simulations in China based on the Bayesian model averaging (BMA) approach. Drought episodes will be detected using the Standardized Precipitation-Evapotranspiration Index (SPEI) in 10 climate divisions of China (Vicente-Serrano et al., 2010). Drought risks will also be quantified using the joint return period of duration and severity calculated by a Bayesian copula approach. The uncertainties in BMA and copula parameters will be addressed within a Bayesian framework, leading to probabilistic hydroclimatic projections and drought return periods. The

Climatic Research Unit (CRU) dataset will be collected to evaluate the BMA-based simulations of precipitation and PET.

This paper is divided into four sections. Section 2 will describe models, algorithms, and datasets used to perform Bayesian multi-model climate simulations and multivariate drought risk projections. Section 3 will systematically evaluate the BMA-based hydroclimate simulations and assess climate change impacts on multidimensional drought risks. Finally, Section 4 will provide a summary and conclusions of this study.



## 2 Data and methods

### 2.1 Two-stage Bayesian multi-model framework

To improve multidimensional drought risk projections, we propose a copula-based two-stage Bayesian multi-model framework, as shown in Fig. 1. The framework explicitly uncovers the cascade of uncertainty in multidimensional drought projections. The first step is to perform ensemble climate simulations, including four climate simulations available in the CORDEX East Asia experiment and one PRECIS simulation over China. The second step is to uncover the uncertainty in model weights (Figs. 1a−e) through the MCMC-based BMA climate simulation, leading to probabilistic hydroclimatic projections in each grid cell (Figs. 1f and 1g). Details of ensemble climate projections are given in Section 2.2. The BMA climate simulations will be compared against the AEM simulations. The third step is to propagate the uncertainty in hydroclimatic projections to the uncertainty in drought projections based on drought indices, leading to multiple scenarios of drought characteristics (i.e., duration and severity), as shown in Figs. 1h and 1j. The drought detection is performed using the 6-month Standardized Precipitation Evapotranspiration Index (SPEI6) over 10 climate divisions in China (see Fig. 2a). The parameters required in the SPEI6 calculation are estimated based on the historical data, and then used to calculate SPEI6 under future climate. The fourth step is to perform the MCMC simulations for copula parameter inference and uncertainty quantification, leading to the uncertainty in the dependence structure of drought variables (Fig. 1i) and thus the uncertainty in return periods of drought episodes (Fig. 1j). The red "whiskers" in Fig. 1j represent the uncertainty of drought characteristics resulting from the climate projection. Details of the copula-based drought risk assessment are provided in Section 2.3.

### 2.2 Bayesian multi-model climate projection

The PRECIS model developed by the UK Hadley Centre, together with four regional climate simulations from CORDEX available for the East Asia domain, were used to assess the changes in hydroclimatic regimes over China. Specifically, the COnsortium for Small-scale MOdelling in CLimate Mode (CCLM) RCM was used to dynamically downscale four Coupled Model Intercomparison Project Phase 5 (CMIP5) GCMs (CNRM-CM5, EC-EARTH, HadGEM2-ES, and MPI-ESM-LR) in the CORDEX East Asia experiment, while the PRECIS model was driven by the HadGEM2-ES (Huang et al., 2018; Rockel et al., 2008; Zhu et al., 2017). All the five simulations have the same horizontal resolution of about $0.44° \times 0.44°$ (~ 50 km) but differ in the model domain. The computational domain of the PRECIS simulation covers a region with $109 \times 88$ horizontal grid points with 19 vertical levels in the atmosphere (see Fig. 2a). In comparison, the CCLM model domain is slightly different with $203 \times 167$ horizontal grid points (see Fig. 2b). The PRECIS climate simulation covers the historical period (1969–2005) and a future period (2006–2099), while the CCLM climate simulation covers the historical period (1951–2005) and a future period (2006–2100). Future simulations for both PRECIS used in this study and CCLM used in the CORDEX East Asia experiment are forced with two emission scenarios, including RCP4.5 and RCP8.5. The 30-year monthly hydroclimatic variables including precipitation and potential evapotranspiration (PET) for the historical (1975−2004) and future (2069−2098) periods are collected from the five climate projections to assess the impact of climate change on hydrological regimes. The



FAO-56 Penman-Monteith Equation was applied to the calculation of PET, which was suggested to yield more realistic estimates than the temperature-only-based Thornthwaite method (Allen et al., 1998; Dai, 2013; Sheffield et al., 2012).

Bayesian model averaging (BMA), as an effective tool of correcting under dispersion in ensemble climate projections, was
used to improve the accuracy of monthly precipitation and PET simulations. Assume that $x = x_1,\ldots, x_K$ signify the ensemble of all considered climate simulations, and $y$ denotes the climate observations. $p_k(y|x_k)$ represents the conditional pdf of $y$ given $x_k$. The probabilistic forecast pdf of $y$ for the multi-model ensemble can be expressed as:

$$p(y \mid x_1 ... x_k) = \sum_{k=1}^{K} w_k p_k(y \mid x_k) \tag{1}$$

where $w_k$ is the BMA weight of model $k$ in the ensemble. The sum of all $w_k$ values is equal to 1 and they are nonnegative,
which reflect how well an individual climate simulation matches the observation in the training period. The conditional pdfs, $p_k(y|x_k)$, are commonly assumed to follow a normal distribution. As a result, the original forecast is usually transformed to the space of normal distribution as:

$$y \mid x_k \sim N(a_k + b_k x_k, \sigma^2) \tag{2}$$

The values for $a_k$ and $b_k$ are member specific, and they are the linear transformation parameters derived by simple linear
regression of observations on each climate simulation in the ensemble, leading to the BMA predictive mean as Eq. (3).

$$E(y \mid x_1 ... x_K) = \sum_{k=1}^{K} w_k (a_k + b_k x_k) \tag{3}$$

BMA has been demonstrated to be a powerful approach to combine an ensemble of climate simulations since it is essentially an "intelligent" weighted average forecast based on the model performance (Raftery et al., 1997; Vrugt, 2016; Vrugt and Robinson, 2007; Wilson et al., 2007; Yang et al., 2011; Zhang et al., 2016). Therefore, BMA was applied to monthly
precipitation and PET for each grid cell with CRU's (Climatic Research Unit) gridded monthly precipitation and PET dataset as reference. The CRU dataset is a global gauge-based climate variable product with a $0.5° \times 0.5°$ grid resolution based on thousands of weather stations (Harris et al., 2014).

The BMA weights and the variance $\sigma^2$ were estimated in this study using the MCMC simulation instead of the EM algorithm. The MCMC simulation has been demonstrated to outperform the EM algorithm, which explicitly samples the posterior
distribution of the BMA parameters for uncovering the uncertainty associated with model weights and thus improving the reliability of climate projections (Duan and Phillips, 2010; Vrugt, ter Braak, et al., 2008). The MCMC-based BMA simulations were performed with uncertainty ranges, enhancing the reliability and credibility of the multi-model climate projections. The MCMC simulation was implemented using the Differential Evolution Adaptive Metropolis (DREAM) algorithm (Vrugt, 2016). According to the Bayes' theorem, the posterior distribution $p(w, \sigma^2|x, y)$ of the BMA weights $w = (w_1,\ldots,w_K)$ and variance $\sigma^2$
given the ensemble simulations $x$ and the observational variable $y$ can be expressed as:



$$p(\boldsymbol{w}, \sigma^2 \mid \boldsymbol{x}, y) = \frac{p(\boldsymbol{w}, \sigma^2) \times p(\boldsymbol{x}, y \mid \boldsymbol{w}, \sigma^2)}{p(\boldsymbol{x}, y)} \tag{4}$$

where $p(\boldsymbol{w}, \sigma^2)$ and $p(\boldsymbol{w}, \sigma^2 | \boldsymbol{x}, y)$ denote the prior and posterior distributions of BMA weights and variance, respectively. $p(\boldsymbol{x}, y | \boldsymbol{w}, \sigma^2) \cong L(\boldsymbol{w}, \sigma^2 | \boldsymbol{x}, y)$ denotes the likelihood function; $p(\boldsymbol{x}, y)$ denotes the evidence that acts as a normalization constant, which can be excluded from the Bayesian analysis in practice. Thus, the formulation of Eq. (4) can be simplified as:

$$p(\boldsymbol{w}, \sigma^2 \mid \boldsymbol{x}, y) \propto p(\boldsymbol{w}, \sigma^2) \times L(\boldsymbol{w}, \sigma^2 \mid \boldsymbol{x}, y) \tag{5}$$

The likelihood function $L(\cdot|\cdot)$ in the MCMC-based BMA projection is commonly logarithmically transformed to Eq. (6) for numerical stability and simplicity, where $n$ represents the number of observations in the training period.

$$\ell(w_1, ..., w_K, \sigma^2 \mid x_1, ..., x_K, y) = \sum_{t=1}^{n} \log\left(\sum_{k=1}^{K} w_k p_k(y \mid x_k)\right) \tag{6}$$

The prior distribution is set as a uniform prior distribution of $\boldsymbol{w} \in [0, 1]^K$ and $\sigma^2 \in [0, 3 \cdot \mathrm{var}(y)]$. The MCMC simulation proceeds
by running multiple Markov chains simultaneously and proposing a candidate point $z_\mathrm{p}$ at each step (Vrugt, 2016; Wang and Wang, 2019). The acceptance or rejection of the candidate depends on the Metropolis acceptance probability:

$$p_{\mathrm{accept}}(z_\mathrm{c} \to z_\mathrm{p}) = \min\left[1, \frac{p(z_\mathrm{p})}{p(z_\mathrm{c})}\right] \tag{7}$$

where $z_\mathrm{c}$ represents the current point, and $p(\cdot)$ represents the probability density. The Markov chain moves to $z_\mathrm{p}$ or not, depending on whether the candidate point is accepted. The convergence of Markov chains indicates that the MCMC evolution
can stop, which is commonly monitored through the multi-chain $\hat{R}$ diagnostic of Gelman and Rubin (1992). Typically, a $\hat{R}$-statistic value below 1.2 indicates that the posterior distribution converges to the stationary distribution. A more detailed description of the MCMC simulation, together with the DREAM algorithm, is available in Vrugt, ter Braak, et al. (2008) and Vrugt (2016).

## 2.3 Copula-based Bayesian multidimensional drought risk projection

Copulas are multivariate cumulative distribution functions that enable us to link the marginal distributions of multiple random variables together to form the joint distribution (Genest et al., 2007; Zhang et al., 2019). The dependence of drought duration and severity detected by the SPEI6 over each of the 10 climate divisions in China (see Fig. 2a) was thus described using copulas in this study, leading to a bivariate return period of drought episodes. The 10 climate divisions were created based on the long-term mean temperature and precipitation as well as the topography in China. Assume that $X = X_1, ..., X_n$ denote $n$
random variables, and $F_1(x_1), ..., F_n(x_n)$ represent their marginal cumulative distribution functions (CDFs), the joint CDF $F(x_1, ..., x_n)$ can be expressed as Eq. (8) according to Sklar's theorem (Sklar, 1959).

$$F(x_1, ..., x_n) = C(F_1(x_1), ..., F_n(x_n)) = C(u_1, ..., u_n) \tag{8}$$



where $C$ is an $n$-dimensional copula, i.e., a joint CDF with uniform margins $(u_1, ..., u_n) \in [0,1]^n$. For the bivariate copula, the joint CDF $p$ of drought severity $X$ and duration $Y$ can be formulated as

$$P(X \leq x, Y \leq y) = C[F(x), G(y)] = p \tag{9}$$

where $F(x) = P(X \leq x)$ and $G(y) = P(Y \leq y)$ are the marginal CDFs of drought severity and duration, respectively. To identify the marginal CDF of drought characteristics, several types of probability distributions, including Nakagami, exponential, Rayleigh, gamma, inverse Gaussian, t location scale, generalized Pareto, Birnbaum-Saunders, extreme value, logistic, lognormal, Weibull, log-logistic, Rician, generalized extreme value, and normal distributions were included as the CDF candidates. The optimal copula families were chosen from a total of 11 candidates, including Independence, Gaussian, Clayton, Frank, Gumbel, Joe, Nelson, Marshal-Olkin, BB1, BB5, and Tawn. Formulas of the copula families are provided in Table 1. Both the marginal CDF and copula families were selected using the Akaike information criterion (AIC). In addition, a randomization strategy (also known as "Jittering") was used to avoid the potentially adverse impact of repeated drought durations on the bivariate analysis (Chambers et al., 2018; De Michele et al., 2013).

The copula parameters were estimated through the MCMC simulation in a Bayesian framework similar to the BMA parameters, leading to the posterior parameter distribution instead of the deterministic maximum likelihood (ML) estimates. Here, the Multivariate Copula Analysis Toolbox (MvCAT) was adopted to infer the MCMC-based copula parameters (Sadegh et al., 2017). The log-likelihood function for copula parameter inference in the MvCAT is expressed as:

$$\ell(\theta \mid \tilde{y}) = -\frac{n}{2}\ln(2\pi) - \frac{n}{2}\ln \sigma^2 - \frac{1}{2}\sigma^{-2}\sum_{i=1}^{n}[\tilde{y}_i - y_i(\theta)]^2 \tag{10}$$

where $\theta$ is the copula parameter set; $n$ denotes the total number of observations; $\sigma$ denotes the standard deviation of measurement error; $\tilde{y}_i$ denotes the empirical joint probability of observation $i$ calculated using Gringorten plotting position (Gringorten, 1963); $y_i(\theta)$ is the joint probability of observation $i$ calculated by the parametric copula with the given parameter $\theta$. Different from the BMA parameters, the prior distributions of copula parameters are drawn using Latin Hypercube Sampling (LHS) which is an efficient sampler and has been widely used for implementing robust MCMC simulations (Huang et al., 2018; Stein, 1987; Vrugt, 2016). The Bayesian inference of copula parameter values requires specifying the initial uncertainty ranges, which are provided in Table 1. More details about the MCMC-based inference of copula parameters can be found in Sadegh et al. (2017).

To project the future drought risks, the joint return period of all the episodes in which drought severity (S) and duration (D) exceed their respective threshold is computed using inclusive probability ("OR" and "AND" case) (Salvadori and De Michele, 2004). The drought return period is commonly proportional to the rarity of drought episodes and the relevant losses, and thus climate-induced drought risks can be evaluated by comparing the return periods under past and future climates. The two cases of bivariate return period can be computed using the copula-based approach as:

$$T_{DS}^{\vee} = \frac{\mu}{1 - F_{DS}(D \leq d, S \leq s)} = \frac{\mu}{1 - C_{DS}(D \leq d, S \leq s, \hat{\theta})} \tag{11}$$



$$T_{DS}^{\wedge} = \frac{\mu}{1 - F_D(D \leq d) - F_S(S \leq s) + F_{DS}(D \leq d, S \leq s)} = \frac{\mu}{1 - F_D(D \leq d) - F_S(S \leq s) + C_{DS}(D \leq d, S \leq s, \hat{\theta})} \qquad (12)$$

where $\mu$ denotes the average inter-arrival time between the occurrences of drought episodes (Zhang et al. 2017). It should be noted that the return period is not deterministic but probabilistic with uncertainty ranges due to the posterior distribution of BMA weights and copula parameters derived from the MCMC simulation.

## 3. Results

### 3.1. Simulation of historical precipitation and PET

Figure 3 presents the MCMC-derived posterior distributions of the BMA weights (a: CNRM-CERFACS-CNRM-CM5; b: ICHEC-EC-EARTH; c: MOHC-HadGEM2-ES; d: MPI-M-MPI-ESM-LR; e: PRECIS) and the variance (f) of each ensemble member for the monthly precipitation at a grid cell (109°E, 36°N) during 1975−2004. The red asterisks in each panel represent the corresponding optimal parameters derived using the EM algorithm. In general, an excellent consistency is observed between the MCMC-derived posterior parameter distributions and the EM estimates within the high-density region. More

importantly, the MCMC simulation provides uncertainty information on the model weights in the BMA prediction, which provides multiple scenarios that can generate equally good climate simulations. In addition, the PRECIS simulation has the largest contribution (nearly 0.5) in the ensemble simulation to reproducing the temporal pattern of monthly precipitation observation.

Figures 4 and 5 display the spatial distributions of the 30-year annual, winter (December-January-February, DJF), and

summer (June-July-August, JJA) mean precipitation and PET, respectively. These spatial distributions are derived from the BMA ensemble simulations and the CRU datasets as well as the absolute model bias generated by the AEM and BMA approaches. In general, the BMA ensemble simulation and the CRU dataset show a similar spatial pattern of the annual, winter, and summer mean precipitation and PET. Compared to the AEM simulations, the BMA ensemble simulations have significantly lower absolute model biases except for the winter mean precipitation over Southeast China. For example, the

AEM simulation tends to underestimate the summer precipitation over Southeast China but overestimate over the Tibetan Plateau (Fig. 4k). Such model bias has been largely reduced by the BMA simulation although dry biases remain over Southeast China (Fig. 4l). The improvement of the BMA simulation upon the AEM simulation is more significant for PET than precipitation. For example, a nearly excellent agreement exists between the BMA-simulated summer PET and the CRU-derived PET over China (Fig. 5l). In comparison, the AEM-simulated summer PET generally has a positive bias of over 0.8

mm/day over Northwest and Southeast China, as well as a negative bias of more than 0.8 mm/day over the Tibetan Plateau. This indicates that the AEM-based projection of drought risks can be largely overestimated over Southeast China based on the climate simulations currently available in the CORDEX East Asia experiment due to the overestimated evapotranspiration and the underestimated precipitation. Such an overestimated risk can be significantly corrected by using the BMA ensemble simulation.





To further evaluate the performance of AEM and BMA in simulating the annual, summer and winter mean precipitation and PET, the Taylor Diagram is used to quantify the consistency between the patterns from two simulations and the CRU observation (Taylor, 2001). The simulated pattern agrees better with observations if the model has a higher correlation and a more consistent standard deviation (SD) with the observation, as well as it lies nearer the "OBS". Figure 6 presents the relative performance of AEM and BMA in simulating the annual, summer, and winter mean precipitation and PET for 10 climate

divisions in China. In general, the BMA simulations have higher CORs than the AEM simulations for both annual and seasonal results. For example, the CORs for the AEM-simulated annual precipitation are below 0.9 for the 10 climate divisions, and nearly all of them are larger than 0.9 for the corresponding BMA simulation (Fig. 6a). The BMA simulation also significantly corrects the variation amplitude of the AEM simulation. For example, the AEM-simulated annual PET for multiple climate divisions has SDs larger than 1.2 while nearly all the BMA-derived SDs lie between 0.8 and 1 (Fig. 6b). In addition, the AEM

simulation has a poor performance of reproducing the winter and summer mean precipitation, as well as the summer mean PET since nearly all of their CORs are smaller than 0.6 (see Figs. 6c, 6e, and 6f). But the corresponding CORs for the BMA simulation are increased for all the climate divisions, especially for the summer mean precipitation (Fig. 6e). Therefore, it is evident that the BMA simulation outperforms the AEM simulation in simulating the annual, winter, and summer precipitation and PET for 10 climate divisions in China. It should also be noted that the BMA simulation does not necessarily outperform

the AEM simulation in terms of the variation amplitude. For example, although the CORs for the BMA-simulated winter mean PET have a slight increase upon the AEM simulation, the consistency of the SDs with observations is reduced for several climate divisions.

    Figures 7 and 8 demonstrate the annual cycles of precipitation and PET, respectively. These cycles are estimated from the AEM and BMA simulations as well as the observational data for 10 climate divisions in China. In general, the curves of the

AEM and BMA simulations well match the annual cycle of the CRU-derive precipitation and PET for 10 climate divisions. However, the AEM simulation generally overestimates the precipitation over the Tibetan Plateau (Divisions 3 and 5) but underestimates the precipitation over the other climate divisions. Such biases can be caused by the convection parameterization schemes used in the RCMs, which misrepresent the diurnal cycle of convective precipitation that is dominant over East China. BMA can largely reduce the bias of precipitation simulation compared to the AEM simulation, especially for Divisions 3 and

5. For example, the AEM-simulated precipitation has a wet bias of more than 2 mm/day for Division 5 and a dry bias of nearly 3 mm/day for Division 8 in July, while the corresponding BMA-simulated biases are reduced to approximately 0.5 and 2.2 mm/day. In addition, the AEM simulations show smaller biases for PET than precipitation, but the biases are significant over several divisions. Specifically, the AEM-simulated PET has a generally negative bias of more than 1 mm/day over the Tibetan Plateau (Divisions 3 and 5) but tends to have a positive bias over Southeast China during the warm season (Divisions 7, 8, 9,

and 10). Such biases can be corrected by the BMA simulation, leading to a nearly excellent agreement between the simulated and observed PET for all the climate divisions. The MCMC-based BMA simulations provide the 95% uncertainty ranges of precipitation and PET due to the sampling of the posterior distributions of climate model weights. Therefore, BMA leads to a better performance than AEM to reproduce the annual cycle of precipitation and PET for 10 climate divisions over China.



## 3.2 Projection of future precipitation and PET

Since the drought occurrence is closely related to precipitation and PET, the BMA-derived projection of future precipitation
and PET is vital for assessing the climate-induced changes in drought risks. Figure 9 presents the absolute and relative changes
in the 30-year annual, winter (DJF), and summer (JJA) precipitation derived from the BMA projection between past
(1975−2004) and future (2069−2098) climates under RCP4.5 and RCP8.5. Note that the presented BMA projections are based
on the "best" BMA weights in the MCMC-derived posterior distribution for better visualization. The most significant drying
appears over northern Xinjiang, the western Tibetan Plateau, and Sichuan Basin, with a considerable reduction in the amount
of annual and seasonal precipitation. The other parts of China are expected to become wet under both scenarios, especially for
southern Xinjiang and the northern Tibetan Plateau under RCP8.5 (Fig. 9j). The percentage changes of annual, summer, and
winter precipitation will not generally exceed 40%. In addition, there are large discrepancies between summer and winter
precipitation changes. For example, the precipitation over southeast coastal areas of China is projected to increase by 1 mm/day
in summer but decrease by 0.4 mm/day in winter under both RCP4.5 and RCP8.5 scenarios. It can be seen that the absolute
change of precipitation in summer is generally larger than that in winter, but the relative change shows little difference.
Although the absolute change magnitude of summer precipitation generally increases from west to east of China, such patterns
are not observed for the relative change. Additionally, the increase of the radiative forcing does not necessarily lead to a
significant change in the projected mean precipitation, such as the summer mean precipitation over the southeast coastal areas.

295       In addition to precipitation, the 30-year annual, winter (DJF), and summer (JJA) mean PET are also compared between
past (1975−2004) and future (2069−2098) climates under RCP4.5 and RCP8.5, as shown in Fig. 10. In general, the annual,
winter, and summer PET are projected to increase under both scenarios, with an absolute magnitude of less than 0.7 mm/day
and a relative magnitude of less than 30%. The absolute increase of PET in summer is remarkably larger than the PET increase
in winter, but the relative increase shows no significant difference between summer and winter PET. For example, the projected
increase in winter PET does not generally exceed 0.15 mm/day in most areas of China under RCP8.5 (Fig. 10e), but the
corresponding increase in summer is generally over 0.3 mm/day and even exceeds 0.6 mm/day in the western Tibetan Plateau,
Sichuan Basin, and Northeast China (Fig. 10f). And the relative increase for both summer and winter PET is generally larger
than 10% under RCP8.5 for most areas in China (Figs. 10k and 10l). In addition, the degree of absolute and relative changes
can also be magnified by the increase in the radiative forcing, especially for the summer mean PET.

## 305   3.3 Multidimensional drought risk assessment

To uncover the interaction of drought characteristics, copulas were used to construct the joint probability distribution between
drought severity and duration detected by the SPEI6 for 10 climate divisions in China during 1975−2004. The MCMC
simulations were performed for parameter inference and uncertainty quantification of the chosen optimal copula family. Figure
11 presents the marginal posterior distribution of copula parameters derived from the MCMC simulation. The red asterisk in
each panel represents the ML estimates derived by the frequentist copula approach. The frequentist copula approach refers to





the use of local optimization algorithms (e.g., the L_BFGS-B method) with initial values to estimate parameters (Yan, 2007). The Marshall-Olkin copula with two parameters was chosen for Divisions 1−3 and 8; the Clayton copula was chosen for Divisions 4−7; the Gumbel copula was chosen for Divisions 9−10. Most of the posterior parameters are well constrained with normal distributions, but some are not, especially for the second parameter $\theta_2$ of the Marshall-Olkin copula (e.g., Figs. 11d

and 11f), with a nearly uniform marginal distribution. Such unconstrained parameter distributions can be due to the limited samples of drought episodes. In addition, there is generally a plausible consistency between the posterior distribution of copula parameters inferred by the MCMC simulation and the ML estimates from the frequentist approach for most copula families, but divergent parameter estimates exist for several copulas (e.g., Figs. 11c and 11e). Such a divergence does not imply that the frequentist copula approach provides unreliable simulations but indicates that the frequentist approach provides only one

plausible estimate. In comparison, the MCMC-derived posterior parameter distribution provides multiple equally good copula simulations, enhancing the reliability of multidimensional drought risk assessments.

To examine the fit quality of copulas, the joint probability derived from the empirical copulas and the parametric copulas are compared against each other, as shown in Fig. 12. The comparisons between the MCMC-based "best" copula and the frequentist copula are distinguished by different colors. The closer the points are to the diagonal in the diagnostic plot, the

better the copula fitting is. In general, both the MCMC-based and frequentist approaches provide plausible copula simulations, especially for Divisions 1 and 9. But the frequentist approach tends to underestimate the joint probability compared to the empirical joint probability. Such an underestimation does not necessarily lead to biased copula simulations but can be potentially risky since the frequentist approach fails to guarantee the global optimization for reproducing the joint distribution of observations.

To further uncover the uncertainty in drought risk assessments, Fig. 13 depicts the return period ("AND" and "OR" case) based on drought duration and severity for Division 3 during 1975−2004, derived from the frequentist and Bayesian copulas. Although Figs. 11e and 12c indicate the difference between the frequentist and Bayesian copula simulations, Fig. 13 generally shows an excellent agreement between the drought return period isolines, indicating that the Bayesian approach uncovers the equifinality of copula simulations. The Bayesian copula also uncovers considerable uncertainty in the drought return period

level, especially for those long ones. For example, for a drought episode with a SPEI6 severity of 17.5 and a return period of 50 years, the drought duration can reach 8 months at least and 13 months at most considering the uncertainty in copula parameters. Therefore, the Bayesian copula improves drought risk assessments through a robust assessment of multidimensional characteristics of drought episodes and associated uncertainties.

### 3.4 Bayesian multi-model drought risk projection

Figure 14 presents the comparison of drought severity and duration detected by the 6-month SPEI between the historical (1975−2004) and future (2069−2098) periods over 10 climate divisions in China. In general, both the drought severity and duration are projected to increase over 10 climate divisions. For example, the median drought durations are approximately 2 months over Divisions 4−6 for the historical period (1975−2004) and are projected to increase to 5 months for the future period





(2069−2098). The increase of the radiative forcing leads to no significant increase in the drought duration, but instead causes

a slight increase in the drought severity. In addition, the projected drought severity and duration show less variability for most climate divisions in China. For example, the drought duration for Division 2 has the interquartile ranges (IQR) of 7 and 2 months for past and future climates, respectively. A remarkable exception is that there is no significant change in the drought duration and severity for Division 3 (i.e., the west central Tibetan Plateau). This does not indicate that the climate-induced drought risks will not increase since the multiple droughts with relatively long duration and high severity (indicated by the

outliers) are projected over the west central Tibetan Plateau.

Figure 15 presents the box-and-whisker plots of the number of drought episodes for the past and future climates over the 10 climate divisions. The thick horizontal bar represents the median value, while the lower and upper edges of the box represent the 25th ($Q_1$) and 75th ($Q_3$) percentile values, respectively. The upper and lower whiskers represent the values of $Q_3 + 1.5 \times$ IQR and $Q_1 − 1.5 \times$ IQR, respectively. The values beyond the end of the whiskers are indicated by outlier points. The

uncertainty in the projected number of future drought episodes results from the posterior distribution of BMA weights inferred by the MCMC simulation. Such an uncertainty is limited for several divisions and thus does not propagate to the uncertainty in the number of drought episodes (e.g., no box and whisker for RCP4.5 in Divisions 7−10). In general, the frequency of drought episodes shows no significant increase but an obvious decrease for certain climate divisions. For example, Division 5 experienced 37 drought episodes during 1975−2004, while the corresponding number of drought occurrence is expected to

decrease to 29 under RCP4.5. In addition, the increase in the radiative forcing tends to increase the frequency of drought occurrence, such as Divisions 1, 2, and 3. The uncertainty was also amplified as the increase in the radiative forcing. For example, the number of drought occurrence over Division 5 lies in the range of 20−28 and 24−40 under RCP4.5 and RCP8.5, respectively.

To further quantify the climate-induced change in drought risks, the return periods ("AND" and "OR" cases) of drought

episodes based on drought duration and severity are assessed for the historical and future periods, as shown in Fig. 16. The historical drought duration and severity were used to construct the parametric copula, which was then used to calculate the return period for each drought episode under past and future climates, leading to the box-and-whisker plots of return period in Figure 16. Due to the increase in the drought duration and severity as shown in Fig. 14, most climate divisions are expected to more than double the drought return period except Division 3. For example, the median return periods of historical drought

episodes over Division 1 are 2.8 and 1.3 years for the "AND" and "OR" cases, respectively. And the corresponding cases of return period are expected to increase by 132% and 183%, respectively, under RCP4.5 for the future period. We also observe that the projected return periods of extreme drought episodes increase from RCP4.5 to RCP8.5 for most climate divisions. For example, quite a few drought episodes with "OR" case return period higher than 10 years are projected for Division 3 under RCP8.5, and such extreme episodes do not occur under RCP4.5. Another typical example is the "AND" case for Division 10,

where the occurrence of the drought return periods higher than 10 years significantly increases from RCP4.5 to RCP8.5. This indicates that the extreme droughts are projected to increase in terms of frequency and the associated risks as the radiative forcing increases.



## 4. Conclusions

In this study, a probabilistic projection of multidimensional drought risks was developed through a copula-based two-stage
Bayesian multi-model framework. An ensemble of five regional climate simulations was used to project future changes in
hydroclimatic regimes over China through an MCMC-based BMA approach. A Bayesian copula approach was also introduced
to explicitly uncover potential interactions of the SPEI-detected drought characteristics and associated uncertainties, thereby
improving the multidimensional drought risk assessment. The proposed Bayesian framework addresses the cascade of
uncertainty in the climate-induced multidimensional drought risk projections. We also examined the performance of arithmetic
ensemble mean (AEM) and BMA simulations in reproducing the historical climate, as well as Bayesian and frequentist copula
approaches used for multidimensional drought simulations.

   The BMA climate simulation results indicate that the MCMC simulation provides not only plausible estimates but also the
uncertainty information on the relative contribution of individual models in the multi-model climate simulation, improving the
reliability of ensemble climate projections. And the PRECIS simulation has a relatively large contribution in the ensemble
climate simulations. The AEM climate simulations based on the current CORDEX East Asia experiments show large biases
in most areas of China. In comparison, the BMA climate simulation can largely improve the simulation of precipitation and
PET, with a higher correlation with observations and smaller biases than the AEM simulation. Both the AEM and BMA climate
simulations can well reproduce the annual cycle of precipitation and PET in China, while AEM shows larger errors and BMA
successfully corrects the errors. The introduced Bayesian copula approach not only provides equally plausible estimates
compared to the frequentist copula approach but also explicitly uncovers the equifinality in the copula simulation. Such an
uncovered equifinality can improve the risk assessment of multidimensional droughts by providing multiple scenarios.

   The MCMC-based BMA climate projections indicate a general increase in future precipitation and PET under RCP4.5 and
RCP8.5 for most areas of China. A considerable decrease in the mean precipitation is also observed in northern Xinjiang, the
western Tibetan Plateau, and Sichuan Basin. The increase in the radiative forcing leads to significant amplification of the PET
increase but does not cause an obvious difference in the precipitation change. The projected changes in future precipitation
and PET cause a general increase in the drought duration and severity with decreasing variability. The drought risks are thus
expected to significantly intensify, with more than doubling the multidimensional return period, even though the occurrence
of drought episodes shows no significant increase and tends to decrease obviously. These findings reveal that China will
experience more frequent extreme droughts, and the associated risks will be elevated due to the increase in the radiative forcing.
The proposed two-stage Bayesian multi-model framework enables a reliable projection of multidimensional drought risks,
which is vital for improving mitigation and preparedness of drought hazards and for enhancing sustainable water resources
management. This framework can be directly applicable to assessing a variety of multidimensional extreme risks, which is
useful for improving the robustness of the risk assessment of extreme events and natural hazards. It should be noted that
although the MCMC-based BMA approach significantly improves the ensemble mean climate simulation, the potential errors
are not completely corrected. It is thus desired to further improve regional climate simulations using the high-resolution



convection-permitting modeling systems in future studies. In addition, the time-invariant BMA weights determined by the historical data in multi-model climate projections may not well represent the nonstationary nature of climate dynamics. Although the underlying uncertainty in the BMA weights was explicitly addressed in this study and previous studies also yielded plausible results (Olson et al., 2016, 2018; Shin et al., 2019; Terando et al., 2012), it is desired to develop nonstationary

frameworks to further improve the credibility of climate projections.

*Data availability.* The CRU dataset was obtained from the CEDA Archive (https://www.ceda.ac.uk/). The CORDEX model outputs were extracted from the ESGF portal (https://esgf.llnl.gov/). The PRECIS model outputs and other related data used in this paper are available in Zhang (2019), doi: 10.17632/n8ckgdy2rr.2.

*Author contribution.* Shuo Wang and Boen Zhang developed the main ideas. Jinxin Zhu performed the climate simulation.

Boen Zhang implemented the computer code and carried out analyses. Boen Zhang and Shuo Wang prepared the manuscript.

*Acknowledgments.* This research was supported by the National Natural Science Foundation of China (Grant No. 51809223), the Hong Kong Research Grants Council Early Career Scheme (Grant No. PP5Z), the Hong Kong Polytechnic University Internal Research Grant (Grant No. YBZ0), and the Hong Kong Polytechnic University Start-up Grant (Grant No. ZE8S).

*Competing interests.* The authors declare that they have no conflict of interest.




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





**List of Figure Captions**

**Figure 1**. Schematic of the two-stage Bayesian multi-model framework for probabilistic multidimensional drought risk projections. The model structure uncertainty consists of five RCM datasets. (a)−(e) denote the model weight uncertainty, i.e., the posterior distributions of BMA weights derived from MCMC simulation. (f) and (g) represent the BMA-derived precipitation and potential evapotranspiration (PET), respectively, as well as their uncertainty ranges. (h) denotes the

uncertainty range of drought index (SPEI used in this study) owing to the uncertainty in the BMA-derived precipitation and PET, leading to a probabilistic delineation of drought episodes. (i) represents the uncertainty information of the dependence structure between drought characteristics (drought duration and severity) derived from the MCMC simulation, leading to the uncertainty in the return period of drought episodes as shown in (j). The red "whiskers" in (j) represent the uncertainty of drought characteristics resulting from the climate projection.

**Figure 2.** (a) The PRECIS model domain with topography and 10 climate divisions including: 1. Cold-temperature and humid zone; 2. Warm-temperature and arid zone; 3. Plateau and semi-arid zone; 4. Warm-temperature and semi-arid zone; 5. Plateau and semi-humid zone; 6. Mid-temperature and humid zone; 7. Warm-temperature and humid zone; 8. North-subtropical and humid zone; 9. Mid-subtropical and humid zone; 10. South-subtropical and humid zone. The 10 climate divisions are generated based on the long-term mean temperature and precipitation as well as the topography in China. (b) The COnsortium for Small-

scale MOdelling in CLimate Mode (CCLM) model domain with topography.

**Figure 3**. Marginal posterior probability distributions of the MCMC-derived BMA weights (a−e) and variance (f) of the individual ensemble members for precipitation. The red asterisks show the corresponding estimates derived using the Expectation-Maximum (EM) algorithm.

**Figure 4**. Spatial patterns of 30-year annual and seasonal mean precipitation (unit: mm/day) generated from the MCMC-based

BMA approach (a, e, i), the CRU observation (b, f, j), the absolute model bias between the AEM simulation and the CRU observation (c, g, k) as well as between the BMA simulation and the CRU observation (d, h, l). DJF stands for December, January, and February; JJA stands for June, July, and August. BMA is the Bayesian model averaging, CRU is the Climatic Research Unit, and AEM is the arithmetic ensemble mean. MCMC is the Markov chain Monte Carlo.

**Figure 5**. Spatial patterns of 30-year annual and seasonal mean potential evapotranspiration (PET) (unit: mm/day) generated

from the MCMC-based BMA approach (a, e, i), the CRU observation (b, f, j), the absolute model bias between the AEM simulation and the CRU observation (c, g, k) as well as between the BMA simulation and the CRU observation (d, h, l).

**Figure 6.** Comparison of precipitation and PET derived from the BMA and AEM simulations for 10 climate divisions in China. (a−b), (c−d), and (e−f) correspond to Annual, DJF, and JJA, respectively. Each point represents the simulation for a climate division, which agrees better with the observation when it has a higher correlation and a more consistent standard

deviation with the observation, as well as it lies nearer the "OBS".



**Figure 7.** Annual cycle of precipitation (units: mm/day) generated from the BMA and AEM simulations as well as the CRU observation over 10 climate divisions in China. The BMA-derived precipitation is shown as the 95% uncertainty ranges owing to the uncertainty in BMA weights derived by the MCMC simulation.

**Figure 8.** Annual cycle of PET (units: mm/day) generated from the BMA and AEM simulations as well as the CRU observation over 10 climate divisions in China. The BMA-derived PET is shown as the 95% uncertainty ranges owing to the uncertainty in BMA weights.

**Figure 9.** Absolute (a−f) and relative (g−l) differences of 30-year annual, winter (DJF), and summer (JJA) mean precipitation between past and future climates. The future precipitation is projected using the BMA approach with the "best" weights in the MCMC-derived posterior distributions.

**Figure 10.** Absolute (a-f) and relative (g-l) differences of 30-year annual, winter (DJF), and summer (JJA) mean PET between past and future climates. The future PET is projected using the BMA approach with the "best" weights in the MCMC-derived posterior distributions.

**Figure 11.** The posterior distributions of copula parameters derived from the MCMC simulation for 10 climate divisions in China. The red asterisk in each panel denotes the maximum likelihood (ML) estimates derived by the frequentist approach.

**Figure 12.** Comparison of the empirical and fitted joint probability for 10 climate divisions in China. The fitted joint probability is separately calculated using copulas inferenced by Bayesian and frequentist approaches, as represented by the red and blue dots, respectively.

**Figure 13.** Comparison of the estimated drought return periods by using Bayesian (blue) and frequentist (red) copulas over Division 2. The 95% uncertainty ranges (grey) are due to the copula parameter uncertainty derived by the MCMC simulation.

**Figure 14.** Box-and-whisker plots of the drought duration and severity for the past and future climates over the 10 climate divisions. The thick black horizontal bars represent the median value, and the lower and upper edges of the box represent the 25th ($Q_1$) and 75th ($Q_3$) percentile values, respectively. The upper and lower whiskers represent the values of $Q_3 + 1.5 \times$ IQR and $Q_1 − 1.5 \times$ IQR, respectively, where IQR denotes the interquartile range that is equal to $Q_3 − Q_1$. The values beyond the end of the whiskers are indicated by outlier points.

**Figure 15.** Box-and-whisker plots of the number of drought episodes for the past and future climates over the 10 climate divisions. The thick black horizontal bars represent the median value, and the lower and upper edges of the box represent the 25th ($Q_1$) and 75th ($Q_3$) percentile values, respectively. The upper and lower whiskers represent the values of $Q_3 + 1.5 \times$ IQR and $Q_1 − 1.5 \times$ IQR, respectively, where IQR denotes the interquartile range that is equal to $Q_3 − Q_1$. The values beyond the end of the whiskers are indicated by outlier points.

**Figure 16.** The return periods of all drought episodes for the past and future climates over the 10 climate divisions. The setting of the box-and-whisker plot is the same as Figure 15. The return periods are calculated by the parametric copula constructed for the historical drought duration and severity that are detected by the 6-month SPEI. The uncertainty in the return period results from the cascade of uncertainty as shown in Fig. 1.



**Figure 1**. Schematic of the two-stage Bayesian multi-model framework for probabilistic multidimensional drought risk projections. The model structure uncertainty consists of five RCM datasets. (a)−(e) denote the model weight uncertainty, i.e., the posterior distributions of BMA weights derived from the MCMC simulation. (f) and (g) represent the BMA-derived precipitation and potential evapotranspiration (PET), respectively, as well as their uncertainty ranges. (h) denotes the uncertainty range of drought index (SPEI used in this study) owing to the uncertainty in the BMA-derived precipitation and PET, leading to a probabilistic delineation of drought episodes. (i) represents the uncertainty information of the dependence structure between drought characteristics (drought duration and severity) derived from the MCMC simulation, leading to the uncertainty in the return period of drought episodes as shown in (j). The red "whiskers" in (j) represent the uncertainty of drought characteristics resulting from the climate projection.


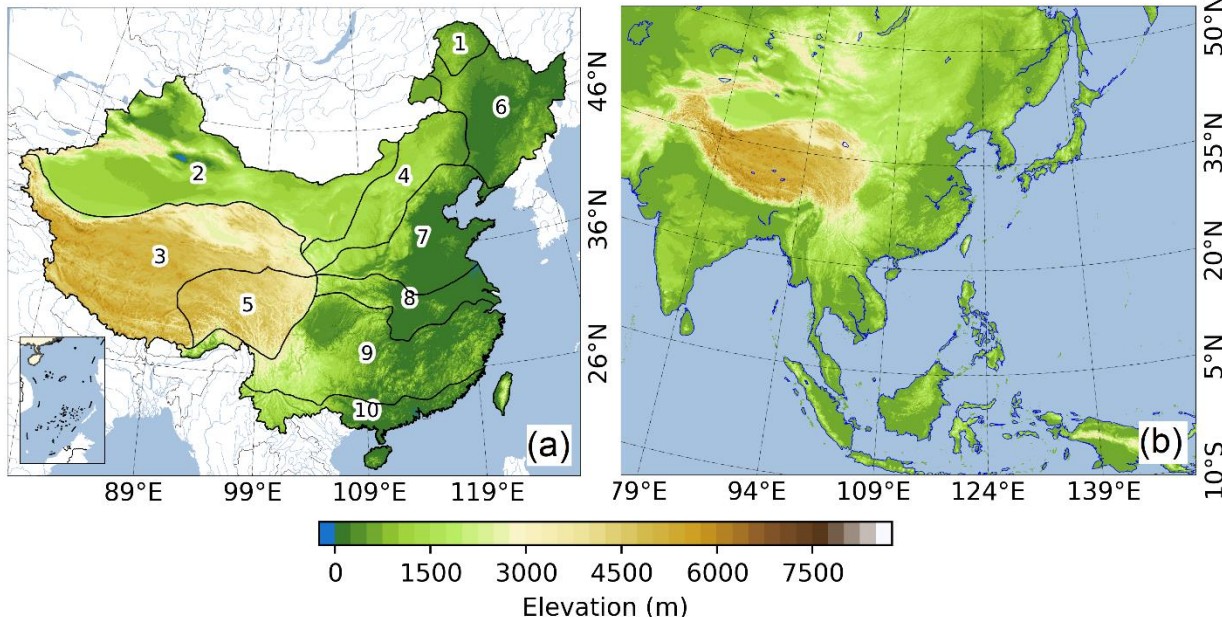

685 **Figure 2.** (a) The PRECIS model domain with topography and 10 climate divisions including: 1. Cold-temperature and humid zone; 2. Warm-temperature and arid zone; 3. Plateau and semi-arid zone; 4. Warm-temperature and semi-arid zone; 5. Plateau and semi-humid zone; 6. Mid-temperature and humid zone; 7. Warm-temperature and humid zone; 8. North-subtropical and humid zone; 9. Mid-subtropical and humid zone; 10. South-subtropical and humid zone. The 10 climate divisions are generated based on the long-term mean temperature and precipitation as well as the topography in China. (b) The COnsortium for Small-690 scale MOdelling in CLimate Mode (CCLM) model domain with topography.

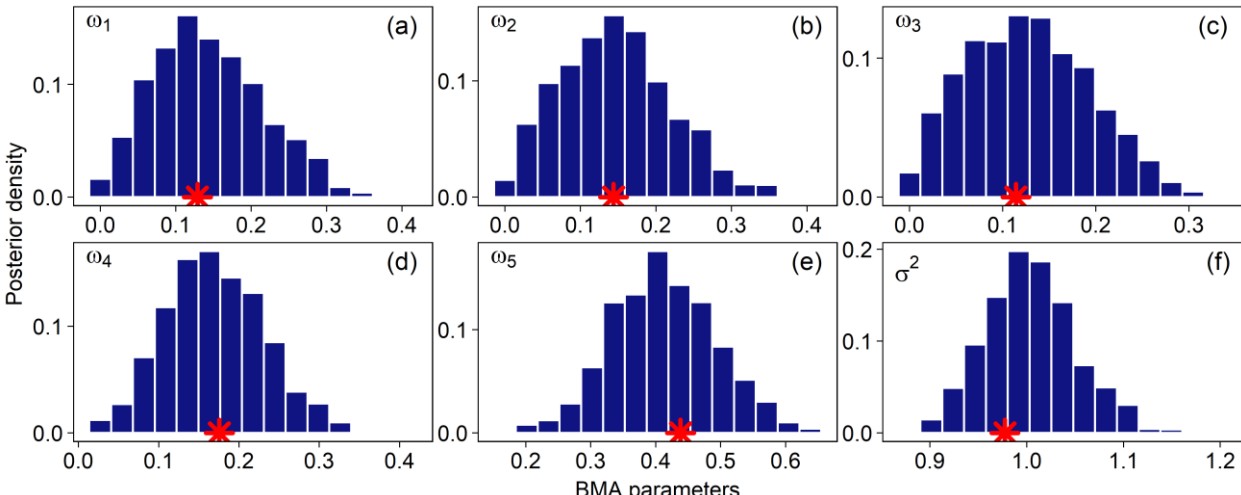

**Figure 3**. Marginal posterior probability distributions of the MCMC-derived BMA weights (a−e) and variance (f) of the individual ensemble members for precipitation. The red asterisks show the corresponding estimates derived using the Expectation-Maximum (EM) algorithm.



**Figure 4**. Spatial patterns of 30-year annual and seasonal mean precipitation (unit: mm/day) generated from the MCMC-based BMA approach (a, e, i), the CRU observation (b, f, j), the absolute model bias between the AEM simulation and the CRU observation (c, g, k) as well as between the BMA simulation and the CRU observation (d, h, l). DJF stands for December, January, and February; JJA stands for June, July, and August. BMA is the Bayesian model averaging, CRU is the Climatic Research Unit, and AEM is the arithmetic ensemble mean. MCMC is the Markov chain Monte Carlo.



**Figure 5**. Spatial patterns of 30-year annual and seasonal mean potential evapotranspiration (PET) (unit: mm/day) generated from the MCMC-based BMA approach (a, e, i), the CRU observation (b, f, j), the absolute model bias between the AEM simulation and the CRU observation (c, g, k) as well as between the BMA simulation and the CRU observation (d, h, l).





**Figure 6.** Comparison of precipitation and PET derived from the BMA and AEM simulations for 10 climate divisions in China. (a−b), (c−d), and (e−f) correspond to Annual, DJF, and JJA, respectively. Each point represents the simulation for a climate division, which agrees better with the observation when it has a higher correlation and a more consistent standard deviation with the observation, as well as it lies nearer the "OBS".

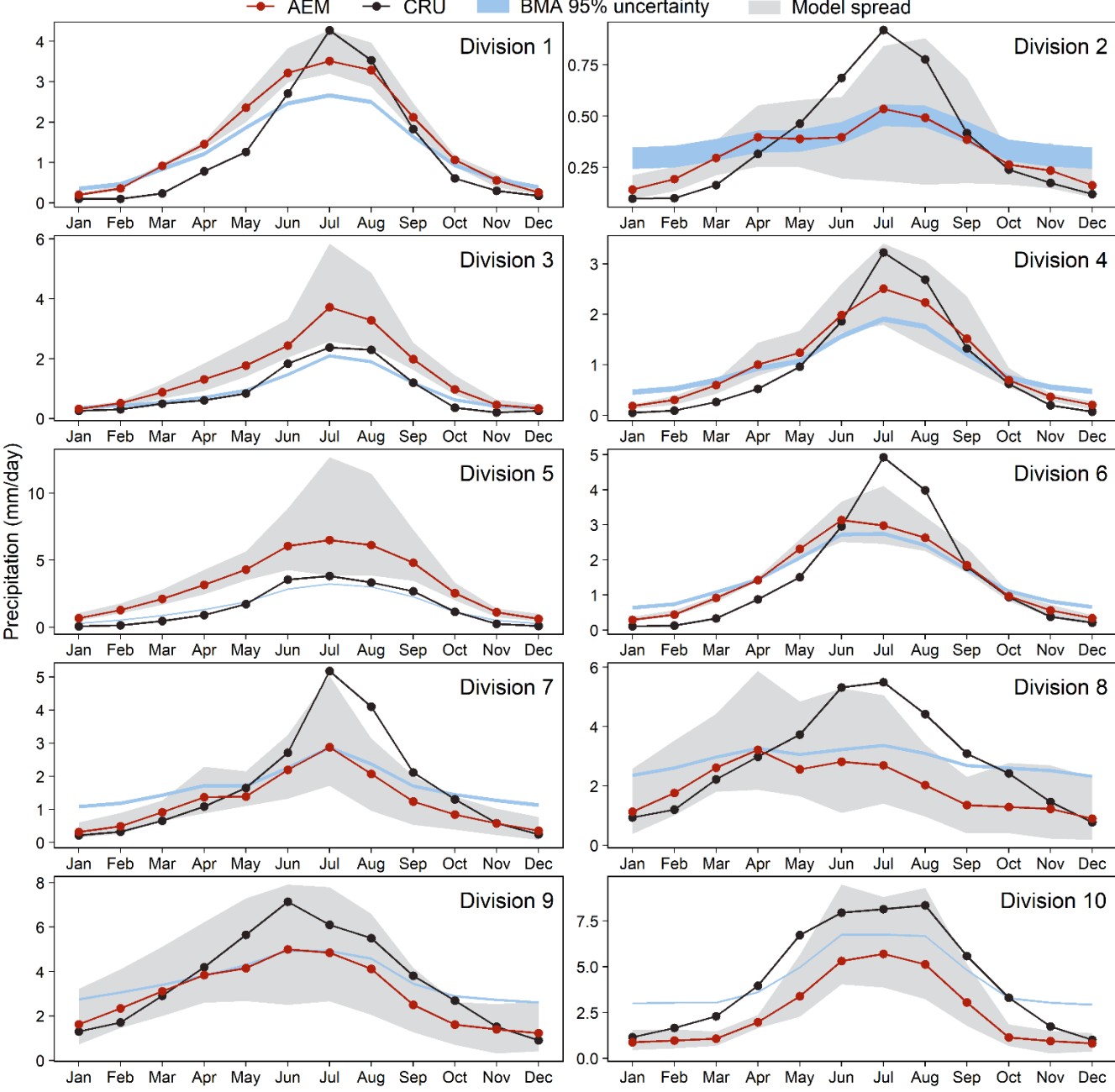

**Figure 7.** Annual cycle of precipitation (units: mm/day) generated from the BMA and AEM simulations as well as the CRU observation over the 10 climate divisions in China. The BMA-derived precipitation is shown as the 95% uncertainty ranges owing to the uncertainty in BMA weights derived by the MCMC simulation.

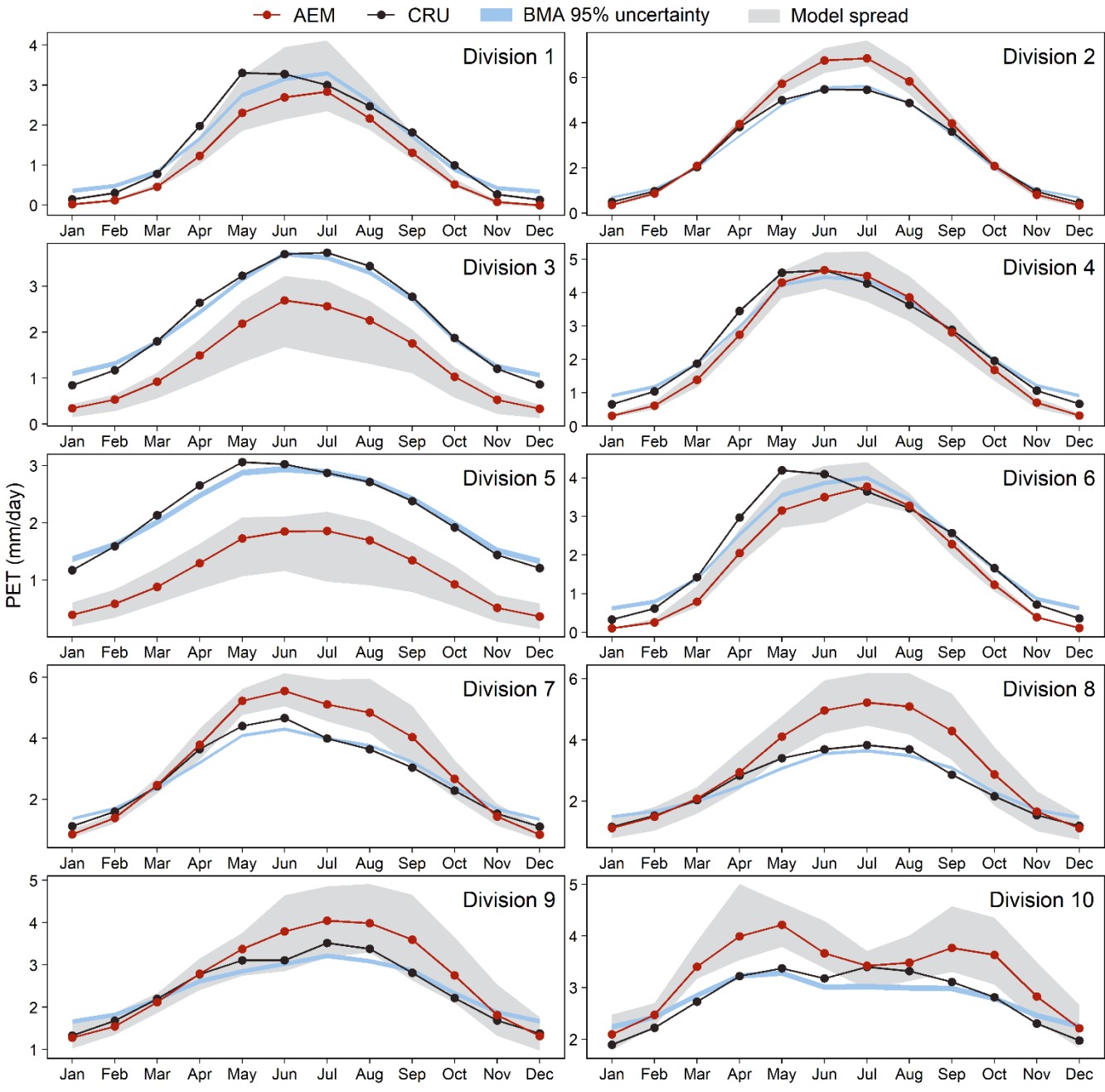

**Figure 8.** Annual cycle of PET (units: mm/day) generated from the BMA and AEM simulations as well as the CRU observation over the 10 climate divisions in China. The BMA-derived PET is shown as the 95% uncertainty ranges owing to the uncertainty in BMA weights.

**Figure 9.** Absolute (a−f) and relative (g−l) differences of 30-year annual, winter (DJF), and summer (JJA) mean precipitation
between past and future climates. The future precipitation is projected using the BMA approach with the "best" weights in the
MCMC-derived posterior distributions.



**Figure 10.** Absolute (a-f) and relative (g-l) differences of 30-year annual, winter (DJF), and summer (JJA) mean PET between past and future climates. The future PET is projected using the BMA approach with the "best" weights in the MCMC-derived

730     posterior distributions.

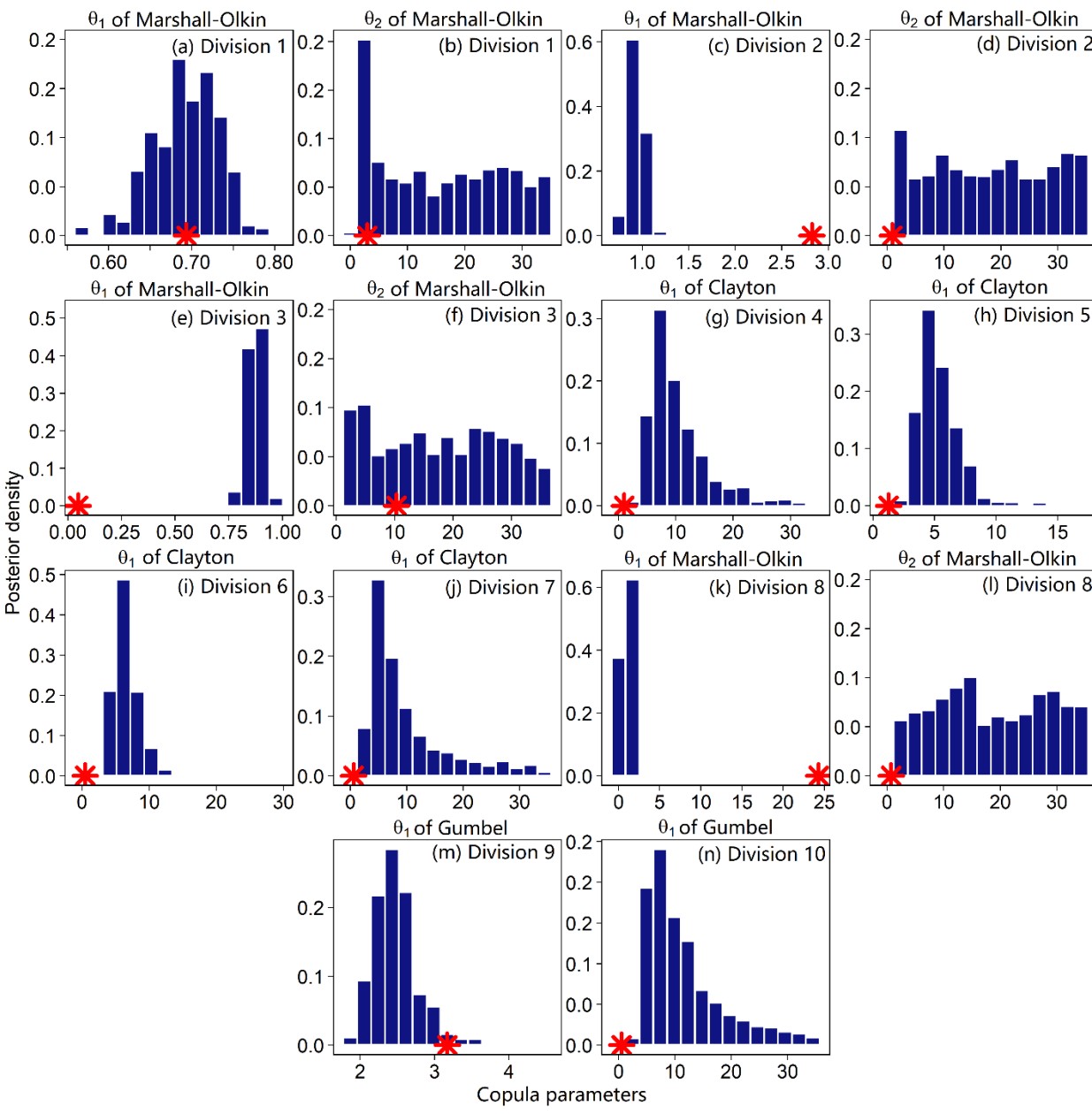

**Figure 11.** The posterior distributions of copula parameters derived from the MCMC simulation for 10 climate divisions in China. The red asterisk in each panel represents the maximum likelihood (ML) estimates derived by the frequentist approach.



735

**Figure 12.** Comparison of the empirical and fitted joint probability for 10 climate divisions in China. The fitted joint probability is separately calculated using copulas inferenced by Bayesian and frequentist approaches, as represented by the red and blue dots, respectively.



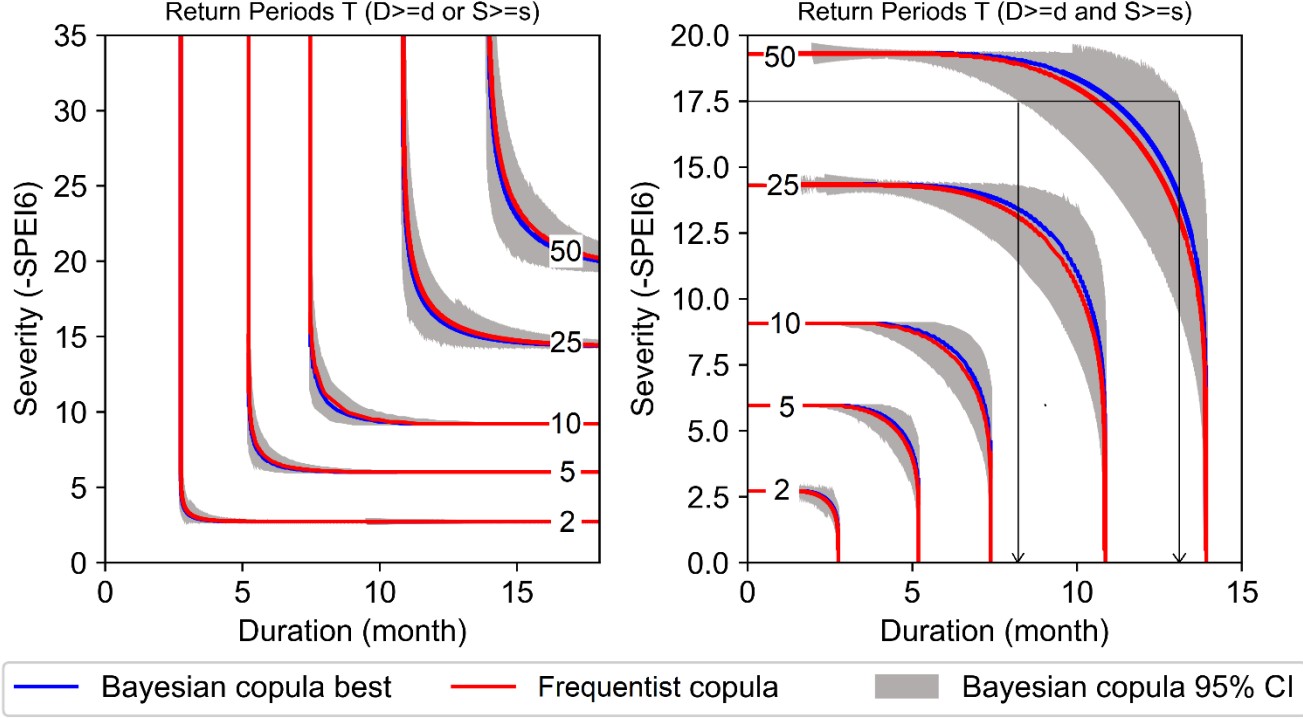

740

**Figure 13.** Comparison of the estimated drought return periods by using Bayesian (blue) and frequentist (red) copulas over Division 2. The 95% uncertainty ranges (grey) are due to the copula parameter uncertainty derived by the MCMC simulation.

745 **Figure 14.** Box-and-whisker plots of the drought duration and severity for the past and future climates over the 10 climate divisions. The thick black horizontal bars represent the median value, and the lower and upper edges of the box represent the $25^{th}$ ($Q_1$) and $75^{th}$ ($Q_3$) percentile values, respectively. The upper and lower whiskers represent the values of $Q_3 + 1.5 \times IQR$ and $Q_1 - 1.5 \times IQR$, respectively, where IQR denotes the interquartile range that is equal to $Q_3 - Q_1$. The values beyond the end of the whiskers are indicated by outlier points.

750



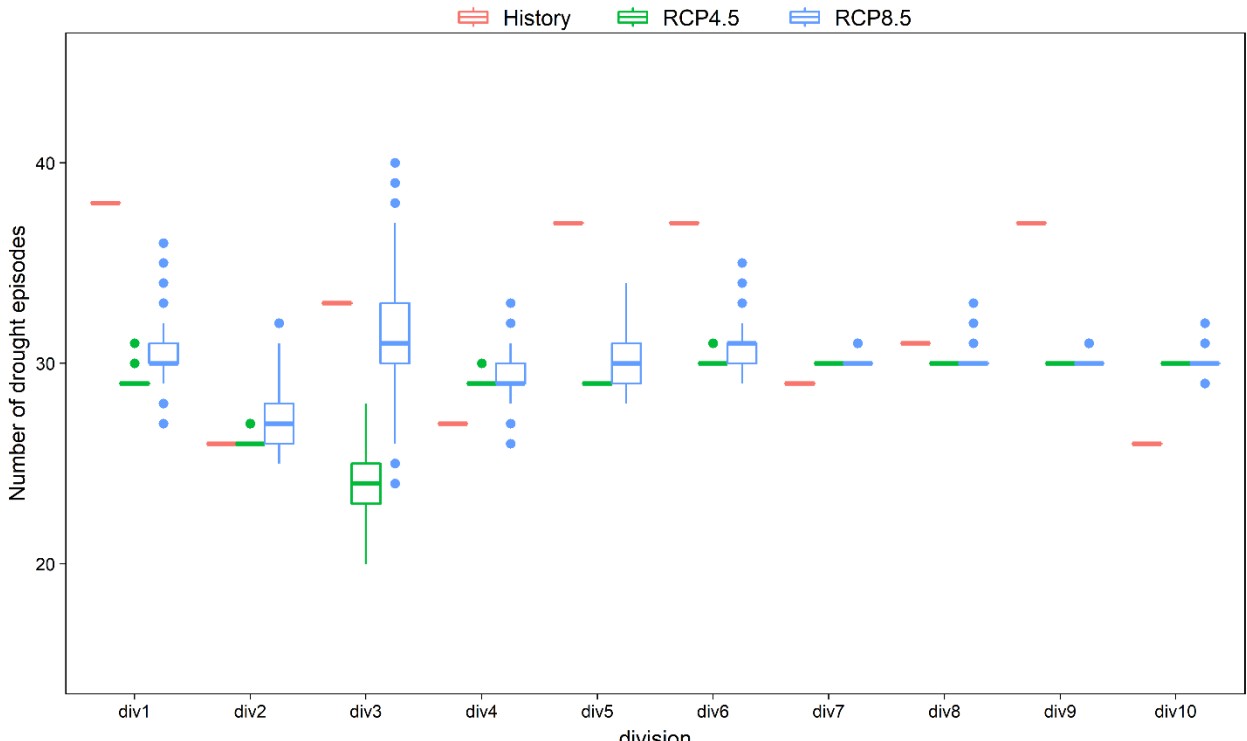

**Figure 15.** Box-and-whisker plots of the number of drought episodes for the past and future climates over the 10 climate divisions. The thick black horizontal bars represent the median value, and the lower and upper edges of the box represent the $25^{th}$ ($Q_1$) and $75^{th}$ ($Q_3$) percentile values, respectively. The upper and lower whiskers represent the values of $Q_3 + 1.5 \times IQR$ and $Q_1 - 1.5 \times IQR$, respectively, where IQR denotes the interquartile range that is equal to $Q_3 - Q_1$. The values beyond the end of the whiskers are indicated by outlier points.

**Figure 16.** The return periods of all drought episodes for the past and future climates over the 10 climate divisions. The setting of the box-and-whisker plot is the same as Figure 15. The return periods are calculated by the parametric copula constructed for the historical drought duration and severity that are detected by the 6-month SPEI. The uncertainty in the return period results from the cascade of uncertainty as shown in Figure 1.





**List of Table Captions**

**Table 1.** Summary of 11 copula families and the corresponding initial parameter uncertainty ranges for the MCMC-based inference

| Name | Mathematical Description for $C(u, v)$ | Parameter Range |
|---|---|---|
| Independence | $uv$ | |
| Gaussian | $\int_{-\infty}^{\phi^{-1}(u)} \int_{-\infty}^{\phi^{-1}(v)} \frac{1}{2\pi\sqrt{1-\theta^2}} \exp\left(\frac{2\theta xy - x^2 - y^2}{2(1-\theta^2)}\right) dx dy$ | $\theta \in [-1,1]$ |
| Clayton | $\max(u^{-\theta} + v^{-\theta} - 1, 0)^{-1/\theta}$ | $\theta \in [-1,35] \backslash 0$ |
| Frank | $-\frac{1}{\theta} \ln\left[1 + \frac{(\exp(-\theta u)-1)(\exp(-\theta v)-1)}{\exp(-\theta)-1}\right]$ | $\theta \in [-35,35] \backslash 0$ |
| Gumbel | $\exp\left\{-\left[(-\ln(u))^{\theta} + (-\ln(v))^{\theta}\right]^{1/\theta}\right\}$ | $\theta \in [1,35]$ |
| Joe | $1 - \left[(1-u)^{\theta} + (1-v)^{\theta} - (1-u)^{\theta}(1-v)^{\theta}\right]^{1/\theta}$ | $\theta \in [1,35]$ |
| Nelson | $-\frac{1}{\theta} \log\left\{1 + \frac{[\exp(-\theta u)-1][\exp(-\theta v)-1]}{\exp(-\theta)-1}\right\}$ | $\theta \in (0,35]$ |
| Marshal-Olkin | $\min[u^{(1-\theta_1)}v, uv^{(1-\theta_2)}]$ | $\theta_1, \theta_2 \in [0,35]$ |
| BB1 | $\left\{1 + \left[(u^{-\theta_1}-1)^{\theta_2} + (v^{-\theta_1}-1)^{\theta_2}\right]^{1/\theta_2}\right\}^{-1/\theta_1}$ | $\theta_1 \in (0,35], \theta_2 \in (1,35]$ |
| BB5 | $\exp\left\{-\left[(-\ln(u))^{\theta_1} + (-\ln(v))^{\theta_1} - \left((-\ln(u))^{-\theta_1\theta_2} + (-\ln(v))^{-\theta_1\theta_2}\right)^{-1/\theta_2}\right]^{1/\theta_1}\right\}$ | $\theta_1 \in [1,35], \theta_2 \in (0,35]$ |
| Tawn | $\exp\left\{\ln(u^{1-\theta_1}) + \ln(v^{1-\theta_2}) - \left[(-\theta_1\ln(u))^{\theta_3} + (-\theta_2\ln(v))^{\theta_3}\right]^{1/\theta_3}\right\}$ | $\theta_1, \theta_2 \in [0,1], \theta_3 \in [1,35]$ |