# Peer review of "A two-stage Bayesian multi-model framework for improving multidimensional drought risk projections over China"

_Hydrology and Earth System Sciences, 2020_

## Referee Comment (RC1) · Anonymous Referee #1 · 4 Sep 2020

In this paper, the authors present a Bayesian multi-model framework used for drought projections analysis. The method is presented, compared (regarding precipitation and potential evapotranspiration) to the ensemble mean when applied over an historic period, and then projections are realised and analysed. I will be honest: I am not familiar with Bayesian and copulas frameworks. Therefore, I will not comment neither the statistical aspects of these methods nor whether this work is really new and sound. I let that to other reviewers. Rather, I will concentrate my review on the climate change aspect as well as whether the presentation and analysis correspond to what is expected in a journal like HESS.

[Figure]

Major remarks:

Presentation of the performance of BMA historical projections: I have the feeling that BMA is slightly oversold. I agree that it performs better than AEM, but cases where it does not perform better are most of the time not mentioned. I spotted some of them in my miscellaneous remarks. I think this is a pity, as understanding and explaining why this is the case could be very interesting. I generally feel that the manuscript lacks a lot of analysis and discussion of the results.

Validity of the method under climate change: the authors analyse the performance of the method under historic data. However, the method is then applied to future climate projections. As the change of climate is rather important in the future, the validity of the method in terms of extrapolation must be questioned. Usually, especially in hydrology under change (see the many "split sample" papers and applications of this method), the historic data is split into two parts, one part being used for optimising the parameters of the model/relation and the second part being used as an independent evaluation. While this does not guarantee that the method might be performing well in the future (since the historic data might be less contrasted that future data), this is a condition that is necessary to satisfy. If the method does not perform well on historic data that is independent, then it is very likely that its behaviour will not be satisfying in the future.

One point that became evident to me when seeing the figures were even not mentioned by the authors. Namely, the two RCPs show very close results with BMA! Check for example Figure 9 (precipitation): if you compare RCP4.5 and RCP8.5 for a specific indicator, the spatial pattern is very similar and magnitudes are also very close. For PET, this is a bit less the case, but still, strongly similar spatial patterns can be observed. Later on, in Figure 14, several boxplots of the drought duration are identical for both RCPs. This is a bit worrying I would say, as I doubt that raw projections also show this behaviour. Could it be that BMA constraints too much projections? To understand that, similar plots without BMA could help.

A second point is seen in Figure 15: some RCPs boxplots (well many in fact) are flat. This means that all projections give the same number of drought episodes. While this could be for an indicator showing small number (e.g. <5), here we are often around 30! This is a bit strange too.

Miscellaneous remarks:

Abstract, line 11: that is not entirely true, as some adaptation strategies are dedicate to tackle flood issues. Line 22-23: which aspect of the risk is expected to double? Frequency? Duration? Intensity? Please be more specific here.

Line 34 and others: there are some surnames in the citations, please correct.

L. 36-38: using ensembles and not restraining the analysis to the mean is already standard in hydrology, including for droughts. Please check for example some examples in HESS: Vidal et al. (2016) and Parajka et al. (2016).

Line 56: while the statement is true, Ramos et al. (2013) is about forecasts, not projections. Extrapolating the conclusion from Ramos to projections is far from trivial I believe. I suggest removing this citation.

L 131: please define the pdf acronym.

L. 232: I quite disagree regarding precipitation. On Figure 4l, we see rather high differences on Eastern China.

L. 250: please define COR.

L. 252-254: this is not so clear for the other ones. See for example the winter PET.

L. 252-254 and lines 260-262 are contradictory

L. 264-265: I rather disagree; there is quite often a factor 2 between OBS and BMA. In addition, some divisions show a peak timing that is not adequately represented.

L. 267-268: nice to finally see some attempt of discussion of the results. However, I

feel we need more: why is this the case? Explain! Please also provide a reference.

L. 269-270: again, I think that the presentation is unfair: the number of opposite behaviour is rather similar from what I see on the figure.

L. 306-313: this is methods, not results. Please move that part in Methods.

L. 312-313: there is no justification why these 3 copulas were chosen for these divisions.

L. 352-354: please remove, this is in the caption already

L. 362: occurrenceS

References: Chambers et al. Is it a book? Please specify the type of work.

Figure 1: we miss a legend in order to have the possibility to understand panels i and j.

Figure 2: while there is only one colour scale, it seems to me that the flat low lands in Eastern China are plotted in different greens. Can you please check?

From Figure 4 onwards: please specify the period of study used for these figures

Figure 4 and 5: we need the AEM plots in order to more easily compare the 3 datasets. That can help us understand the spatial differences, as here for example we have a very unclear idea of of far/close AEM and BMA are from each other.

Figure 6: why there are no line around OBS to help use assess the error, as in classical Taylor diagrams? That would help a lot.

References:

Parajka, J., Blaschke, A. P., Blöschl, G., Haslinger, K., Hepp, G., Laaha, G., Schöner, W., Trautvetter, H., Viglione, A., and Zessner, M.: Uncertainty contributions to low-flow projections in Austria, Hydrol. Earth Syst. Sci., 20, 2085–2101, https://doi.org/10.5194/hess-20-2085-2016, 2016.

Vidal, J.-P., Hingray, B., Magand, C., Sauquet, E., and Ducharne, A.: Hierarchy of climate and hydrological uncertainties in transient low-flow projections, Hydrol. Earth Syst. Sci., 20, 3651–3672, https://doi.org/10.5194/hess-20-3651-2016, 2016.

---

## Referee Comment (RC2) · Anonymous Referee #2 · 28 Sep 2020

Review of " A two-stage Bayesian multi-model framework for improving multidimensional drought risk projections over China" by Zhang et al. The paper applies a Bayesian framework to assess the uncertainty in future projections of drought hazards in China. From reading the abstract and introduction, I am still unsure as too exactly what is the motivation of the study, and what the main hypothesis is. This needs to be made clearer to the reader. The main findings are that the frequency of drought event decreases, but the severity of the events increase. Further they claim that the method can detect and correct for uncertainties in the underlying RCM. I do find the results interesting, and the finding that the drought severity increases is important, but it needs to be further tested for significance and robustness. I also think that the paper reads

more like a technical than a scientific paper, and this needs to be improved. I think that the paper needs a substantial revision before it can be published. Major comments The paper is very technical, and the main point is to apply the Bayesian framework on drought estimations on regional climate model output. I would argue that the uncertainties in both using regional climate modelling and the SPEI are highly uncertain in their underlying assumptions, so trying to correct any errors in the output is almost impossible. It is also a risk that sharpening the results leads to a wrong conclusion, since the methods applied might be completely off. A thorough testing of the methodology on for example reanalysis data would be one way of testing the robustness. The use of SPEI puzzles me, since I do not see how the values can be derived. They are not what is usually found in literature. Usually negative values denote drought conditions, and the values are within a few standard deviations, values outside the range +/-2 is usually considered very wet/dry. Please explain this more. Further, drought indices do not necessarily indicate drought conditions. Each location has their own sensitivity, so the risk of drought should be considered in a study to say something about a severe event, this is captured in a comment below. Thirdly, and most importantly, does the Bayesian framework bring any new light on this topic? What do the raw RCM results say in terms of increase severe drought indices? Are those results too muddled to draw any real conclusions from them? Is that because of a lack of precision in the modelling or because the uncertainties are so vast that it is really difficult to quantify these changes? Please provide these details to motivate why this methodology is necessary for this specific problem. The presentation can be improved. As a reader I am still confused as what the motivation really is behind the study. The numbers of figures are too many and not always relevant. I would like to see more skill assessments and less descriptive figures. Other comments 1. L70 Calculating the drought return period does not necessary quantify the drought risk, rather the drought hazard. In order to get the risk you need to also take into consideration the impact.

2. Figure 1 contains a lot of information and is very difficult to decipher without first having read the paper. I know you are trying to show the work flow, but I would suggest

to remake the figure to make it more schematic, with a few examples illustrating the stages.

3. Figure 3. Please include the RCM names in the figure caption. Also, please use the same scale on the x-axis to make the interpretation easier.

4. L245-262 and figure 6. In the comparison between AEM and BMA in terms of bias and correlation, the authors suggests that BMA outperforms AEM. I do not find that so evident. The correlation generally increases, but the standard deviation is generally worsened. This is noted by the authors, but I would suggest a more thorough analysis of this.

5. L 263-278, Figures 7-8. I would agree with eh authors that PET is generally im- proved with BMA, but I do not find the results for precipitation do so. For many regions BMA seems to smoothen the annual cycle of precipitation by increasing the winter pre- cipitation and decreasing the summer precip. Also, the spread is reduced to a very thin band, which is not really what you want, This is not decreasing the uncertainty, this is being to over-confident with your technique.

6. Figure 9, the bars under each section (a-f,g-l) only needs to be given once

7. Section 3.2 L280-302. You show here the BMA-derived projections, but how do they differ from just using the ensemble mean of the RCM? Are the projected changes significantly different, and if so, why is that?

8. L330-339. I do not understand figure 13. I see the point that you want to compare the copula estimations and their uncertainty, but I do not understand how you can get the SPEI values and their estimated return period. An SPEI value of SPEI of 17.5 is impossible. The value of SPEI. or any standardised index, translates to the number of standard deviations away from the mean, where 1 is one standard deviation away from normal. Or am I misinterpreting the figure? If so, please help me interpret the figure. Same goes for figure 14

9. L389. The authors claims that the reliability of the RCM output is improved, but I see no sign of that ever tested? I would suggest to add reliability diagrams to test this hypothesis properly

10. Conclusions. The authors claim that BMA can "successfully correct" the errors. I disagree with this categorical statement. Some aspects are improved, but others are worsened. This should be clearer.

---

## Author Comment (AC1) · 2 Oct 2020

**Responses to Reviewer #1**

We are grateful to reviewer #1 for his/her constructive comments and suggestions which are helpful to improve the quality of our manuscript. And we will make great efforts to address all the comments, with the details explained as follows.

General comment: Presentation of the performance of BMA historical projections: I have the feeling that BMA is slightly oversold. I agree that it performs better than AEM, but cases where it does not perform better are most of the time not mentioned. I spotted some of them in my miscellaneous remarks. I think this is a pity, as understanding and explaining why this is the case could be very interesting. I generally feel that the manuscript lacks a lot of analysis and discussion of the results.

**Response:** According to the reviewer's constructive comments, we will revise the manuscript with additional explanations and discussions. Our detailed responses to the reviewer's comments are provided as follows.

Comment #1: Validity of the method under climate change: the authors analyse the performance of the method under historic data. However, the method is then applied to future climate projections. As the change of climate is rather important in the future, the validity of the method in terms of extrapolation must be questioned. Usually, especially in hydrology under change (see the many "split sample" papers and applications of this method), the historic data is split into two parts, one part being used for optimising the parameters of the model/relation and the second part being used as an independent evaluation. While this does not guarantee that the method might be performing well in the future (since the historic data might be less contrasted that future data), this is a condition that is necessary to satisfy. If the method does not perform well on historic data that is independent, then it is very likely that its behaviour will not be satisfying in the future.

**Response:** To address the reviewer's comment, a split-sample test was performed to evaluate the robustness of BMA weights. The BMA climate simulations were calibrated during the 20-year period from 1975 to 1994, and then validated during the 10-year period from 1995 to 2004 by comparing against the CRU observations. Figures R1 and R2 (as shown below) depict the spatial patterns of absolute model biases of annual and seasonal precipitation and PET, respectively, generated from the BMA and ensemble mean (AEM) simulations. Results show that there is no significant difference between model biases in the calibration (1975–1994) and validation (1995–2004) periods for the BMA ensemble simulation. The BMA approach significantly reduces the model biases in the calibration and validation periods except for the winter mean precipitation over Southeast China, which has a significant wet bias (Figures R1e and R1f). Although the BMA-simulated summer mean precipitation has a dry bias, it has been largely reduced compared against the AEM simulation. We also observe that the BMA-simulated PET is highly consistent with the CRU PET observations for the calibration and validation periods. This indicates that the robustness of the BMA climate simulations is acceptable for the study area and can lead to reasonable climate projections.

To better address the reviewer's comment, we will add the abovementioned discussions in the revised version of the manuscript and will also investigate the cause of the wet bias in the BMA-simulated winter precipitation.

---

## Author Comment (AC2) · 2 Oct 2020

**Responses to Reviewer #2**

We are grateful to reviewer #2 for his/her constructive comments and suggestions, which are helpful to improve the quality of our manuscript. And we will make great efforts to address all the comments, with the details explained as follows.

*General comment: The paper applies a Bayesian framework to assess the uncertainty in future projections of drought hazards in China. From reading the abstract and introduction, I am still unsure as too exactly what is the motivation of the study, and what the main hypothesis is. This needs to be made clearer to the reader. The main findings are that the frequency of drought event decreases, but the severity of the events increase. Further they claim that the method can detect and correct for uncertainties in the underlying RCM. I do find the results interesting, and the finding that the drought severity increases is important, but it needs to be further tested for significance and robustness. I also think that the paper reads more like a technical than a scientific paper, and this needs to be improved. I think that the paper needs a substantial revision before it can be published.*

*Response:* To address the reviewer's comment, we will clarify the motivation and the main hypothesis in the Introduction section. To test the significance and robustness of the BMA technique, we will compare the AEM- and BMA-simulated hydroclimatic regimes and droughts through quantitative evaluation indices. In addition, we will add more explanations on the results and discussions on the underlying mechanism in the revised version of the manuscript, in attempt to provide new insights into the climate-induced drought risks.

*Comment #1: The paper is very technical, and the main point is to apply the Bayesian framework on drought estimations on regional climate model output. I would argue that the uncertainties in both using regional climate modelling and the SPEI are highly uncertain in their underlying assumptions, so trying to correct any errors in the output is almost impossible. It is also a risk that sharpening the results leads to a wrong conclusion, since the methods applied might be completely off. A thorough testing of the methodology on for example reanalysis data would be one way of testing the robustness. The use of SPEI puzzles me, since I do not see how the values can be derived. They are not what is usually found in literature. Usually negative values denote drought conditions, and the values are within a few standard deviations, values outside the range +/-2 is usually considered very wet/dry. Please explain this more.*

*Response:* To address the reviewer's comment, we will use a split-sample test to evaluate the robustness of the methodology in simulating hydroclimatic regimes and droughts. In addition, we will provide more details on the drought index (i.e., SPEI) in the revised version of the manuscript.

*Comment #2: Drought indices do not necessarily indicate drought conditions. Each location has their own sensitivity, so the risk of drought should be considered in a study to say something about a severe event, this is captured in a comment below.*

*Response:* We agree that drought indices do not necessarily indicate drought conditions. According to the reviewer's comment, we will change the term "drought risk" to "drought hazard".

*Comment #3: Does the Bayesian framework bring any new light on this topic? What do the raw RCM results say in terms of increase severe drought indices? Are those results too muddled to draw any real conclusions from them? Is that because of a lack of precision in the modelling or because the uncertainties are so vast that it is really difficult to quantify these changes? Please provide these details to motivate why this methodology is necessary for this specific problem. The presentation can be improved. As a reader I am still confused as what the motivation really is behind the study. The numbers of figures are too many and not always relevant. I would like to see more skill assessments and less descriptive figures.*

*Response:* According to the reviewer's comment, we will assess the hydroclimatic regimes and droughts generated from the Bayesian framework by comparing against the previous methods, such as the ensemble mean simulation. We will also discuss the deficiency of the previous methods used to simulate droughts.

*Comment #4: L70 Calculating the drought return period does not necessary quantify the drought risk, rather the drought hazard. In order to get the risk you need to also take into consideration the impact.*

*Response:* According to the reviewer's comment, we will change the term "drought risk" to "drought hazard".

*Comment #5: Figure 1 contains a lot of information and is very difficult to decipher without first having read the paper. I know you are trying to show the work flow, but I would suggest to remake the figure to make it more schematic, with a few examples illustrating the stages.*

*Response:* According to the reviewer's comment, Figure 1 will be revised for better presenting the workflow.

*Comment #6: Figure 3. Please include the RCM names in the figure caption. Also, please use the same scale on the x-axis to make the interpretation easier.*

*Response:* Figure 3 will be revised according to the reviewer's suggestion.

*Comment #7: L245-262 and figure 6. In the comparison between AEM and BMA in terms of bias and correlation, the authors suggests that BMA outperforms AEM. I do not find that so evident. The correlation generally increases, but the standard deviation is generally worsened. This is noted by the authors, but I would suggest a more thorough analysis of this.*

*Response:* To address the reviewer's comment, we will provide a through comparison on the performance of the AEM and BMA simulations, and then we will revise the irrelevant statements.

*Comment #8: L 263-278, Figures 7-8. I would agree with eh authors that PET is generally improved with BMA, but I do not find the results for precipitation do so. For many regions BMA seems to smoothen the annual cycle of precipitation by increasing the winter precipitation and decreasing the summer precip. Also, the spread is reduced to a very thin band, which is not really what you want, This is not decreasing the uncertainty, this is being to over-confident with your*

*technique.*

***Response:*** We agree that the BMA technique does not lead to an all-round enhancement upon the AEM approach in simulating precipitation. Consequently, we will use a more appropriate BMA method, such as the copula-based BMA technique proposed by Madadgar and Moradkhani (2014), to improve the reliability of precipitation simulations.

**Reference:**

Madadgar, S., and Moradkhani, H.: Improved Bayesian multimodeling: Integration of copulas and Bayesian model averaging, Water Resour. Res., 50, 9586–9603, doi:10.1002/2014WR015965, 2014.

*Comment #9: Figure 9, the bars under each section (a-f,g-l) only needs to be given once*

***Response:*** Figure 9 will be revised according to the reviewer's suggestion.

*Comment #10: Section 3.2 L280-302. You show here the BMA-derived projections, but how do they differ from just using the ensemble mean of the RCM? Are the projected changes significantly different, and if so, why is that?*

***Response:*** To address the reviewer's comment, we will compare the drought projections generated from the BMA technique and the ensemble mean simulation, and then we will also analyze the underlying reason on the difference.

*Comment #11: L330-339. I do not understand figure 13. I see the point that you want to compare the copula estimations and their uncertainty, but I do not understand how you can get the SPEI values and their estimated return period. An SPEI value of SPEI of 17.5 is impossible. The value of SPEI. or any standardised index, translates to the number of standard deviations away from the mean, where 1 is one standard deviation away from normal. Or am I misinterpreting the figure? If so, please help me interpret the figure. Same goes for figure 14*

***Response:*** We regret for the unclear statement on the calculation of drought characteristics. For better clarification, we will provide detailed statement on the calculation of drought characteristics and the return period. We will also provide more detailed descriptions on Figures 13 and 14.

*Comment #12: L389. The authors claims that the reliability of the RCM output is improved, but I see no sign of that ever tested? I would suggest to add reliability diagrams to test this hypothesis properly.*

***Response:*** To address the reviewer's comment, we will use reliability diagrams and quantitative indices to evaluate the reliability of the BMA-based climate simulations.

*Comment #13: Conclusions. The authors claim that BMA can "successfully correct" the errors. I disagree with this categorical statement. Some aspects are improved, but others are worsened. This should be clearer.*

*Response:* According to the reviewer's suggestion, the section of Conclusions will be revised for better clarification.